# Clinical Utility of Extracorporeal Shock Wave Therapy on Hypertrophic Scars of the Hand Caused by Burn Injury: A Prospective, Randomized, Double-Blinded Study

**DOI:** 10.3390/jcm9051376

**Published:** 2020-05-07

**Authors:** So Young Joo, Seung Yeol Lee, Yoon Soo Cho, Cheong Hoon Seo

**Affiliations:** 1Department of Rehabilitation Medicine, Hangang Sacred Heart Hospital, College of Medicine Hallym University, Seoul 07247, Korea; anyany98@naver.com (S.Y.J.); hamays@hanmail.net (Y.S.C.); 2Department of Physical Medicine and Rehabilitation, College of Medicine, Soonchunhyang University Hospital, Bucheon 14584, Korea; shouletz@gmail.com

**Keywords:** extracorporeal shock wave therapy, hypertrophic scar, burn, hand function

## Abstract

Postburn hypertrophic scarring is a common complication in burn injuries to the hands, often associated with impaired hand function. We evaluated the effects of extracorporeal shock wave therapy (ESWT), compared to a sham stimulation therapy, on hypertrophic scars of the hand caused by burn injury and investigated its effects on hand function. This was a double-blinded, randomized, controlled trial of 48 patients with a burn to their dominant right hand. The parameters of ESWT were as follows: energy flux density, 0.05–0.30 mJ/mm^2^; frequency, 4 Hz; 1000 to 2000 impulses per treatment; four treatments, once a week for four weeks. The outcomes measured were as follows: a 10-point visual analogue scale pain score; Vancouver scar scale for scar vascularity, height, pliability and pigmentation; ultrasound measurement of scar thickness; Jebsen−Taylor hand function test; grip strength; Perdue pegboard test; and the Michigan hand outcomes questionnaire. The change in the score from baseline to post-treatment was compared between the two groups. ESWT improved the pain score (*p* = 0.001), scar thickness (*p* = 0.018), scar vascularity (*p* = 0.0015), and improved hand function (simulated card-turning, *p* = 0.02; picking up small objects, *p* = 0.004). The other measured outcomes were not different between the two groups. ESWT is effective in decreasing pain, suppressing hypertrophic scarring, and improving hand function.

## 1. Introduction

Hypertrophic scarring is a common complication after a burn. Hypertrophic scars result from excessive tissue formation during the wound healing process [1]. The mechanisms underlying hypertrophic scar formation include exaggerated inflammation, prolonged re-epithelialization, excessive extracellular matrix production, and reduced apoptosis [2]. Both types of hypertrophic scars, those which are raised and inflexible [3] and those which are characterized by traction on surrounding tissues, can result in functional limitations [4]. Common complications after burn injuries to the hands include decreased hand performance, sensory impairment, and scar contracture [5]. Hand complications such as these may impair performance of the activities of daily life. Nonsurgical treatments of postburn hypertrophic scars include intralesional corticosteroid injection, laser therapy, compression therapy, and silicone therapy. The outcomes of these nonsurgical treatments of hypertrophic scars on the hands, however, have not yielded satisfactory outcomes. Recently, there has been increased focus on shortening the healing time and improving hand function to reduce the risk of hypertrophic scarring.

Extracorporeal shock wave therapy (ESWT) is effective in the treatment of orthopedic diseases and neuropathic pain and has also been used in regenerative medicine of the skin [6], with neoangiogenesis and anti-inflammatory effects having been reported [7]. The regeneration effects of ESWT have been demonstrated in improving the healing of burn wounds [8,9] as well as in decreasing hypertrophic scarring after a burn [10]. ESWT may also improve the pliability and the subjectively evaluated appearance of postburn scar contracture [11]. Studies in our own institution found ESWT to be useful for burn-associated pain and pruritus and demonstrated that ESWT can suppress the hypertrophic scarring. The anti-fibrotic effects of ESWT can likely be attributed to its molecular effect on hypertrophic scars [1,12,13]. However, the effects of ESWT on hypertrophic scarring of the hand and on hand function have not been evaluated to date. Therefore, this research aimed to evaluate the effectiveness of ESWT on hypertrophic scars and on the hand functions affected by such scarring. 

## 2. Experimental Section

### 2.1. Methods

#### Study Design and Statement of Ethics 

This was a double-blinded, randomized, controlled trial of 48 patients (45 male, 3 female) recruited from the Department of Rehabilitation Medicine, Hangang Sacred Heart Hospital, Korea, between November 2019 and April 2020. Our study was approved by the Ethics Committee of the Hangang Sacred Heart Hospital (HG2018-047) and registered in ClinicalTrials (NCT04138355). All patients provided written informed consent. 

### 2.2. Study Group

Each of the 48 patients enrolled in the study were ≥18 years old and had sustained a deep partial-thickness (second-degree) burn or a full thickness (third-degree) burn involving only the right hand, which had been treated with a split-thickness skin graft (STSG) after the thermal injury, <6 months prior to enrollment. At the time of the study, all the patients were in the re-epithelialization phase of wound healing and had been transferred to the rehabilitation department, after acute burn treatment, to improve hand and wrist stiffness due to painful retracting scarring. The criteria for exclusion from the study were as follows: left-hand dominance; musculoskeletal diseases (fracture, amputation, rheumatoid arthritis, and degenerative joint diseases) of the right dominant hand; pregnancy; or potential for additional damage to the skin if exposed to ESWT and conventional occupational therapy. 

The first 50 burn patients who met our inclusion/exclusion criteria were randomly allocated, using a computer program, to either the ESWT (n = 25) group or to the sham (n = 25) group. While all participants in the sham group completed the trial protocol, two patients in the ESWT group dropped out of the study because they did not want to undergo serial evaluations. Thus, 23 patients were ultimately included in the ESWT group, and 25 were included in the sham group (Figure 1). 

### 2.3. Intervention

Patients in both groups received standard rehabilitation treatment for burn injuries to the hands, including medication, scar lubrication, massage therapy to the scars, and occupational therapy to improve hand function. Occupational therapy treatment consisted of 20 sessions (30 min per day, five days a week) for four weeks. Every effort was made to provide the same exercise rehabilitation for both groups and to adapt the level of difficulty of the exercises prescribed to each patient’s performance. Patients received one ESWT or sham treatment session per week for four weeks. 

Patients in the ESWT group were asked to select the most hypertrophic and retracting scar on their dominant right hand for treatment. ESWT was conducted using the Duolith SD-1 device (StorzMedical, Tägerwilen, Switzerland), with an electromagnetic cylindrical coil source used to focus the shock wave (Figure 2). ESWT was performed around the primary treatment site, at an intensity of 100 impulses/cm^2^, an energy flux density (EFD) of 0.05 to 0.30 mJ/mm^2^, and a frequency of 4 Hz. Between 1000 to 2000 impulses were administered per session, for four sessions held at one week intervals. In the sham group, the same shock wave equipment, with the same-shaped adapter, as in the experimental group was used, but with no energy emitted (Figure 3).

### 2.4. Outcome Measures

In order to evaluate the effect of ESWT, we compared the change in the severity of pain, scar thickness, and hand function between the ESWT and sham groups, from baseline measures taken immediately before the intervention and measures taken immediately after session 4. The 10-point visual analogue scale (VAS) was used to measure self-reported pain severity, with ratings from 0 (no pain) to 10 (unbearable pain). The Vancouver scar scale (VSS), an observer-dependent scale of the macroscopic appearance of scarring, was used to measure changes in the scar over the four week period of treatment [14]. The VSS consists of four parameters (vascularity, height, pliability, and pigmentation), scored in total on a 14-point scale, with a higher score indicative of a better outcome. The thickness of the scar was objectively quantified by ultrasound (128 BW1 US system, Medison, Korea). Grip and pinch strength were quantified using a hand-held dynamometer (Lafayette Instrument, USA). The Michigan hand outcomes questionnaire (MHQ) was used to assess a patient’s perception of hand function on a scale of 0–100, with a higher score indicative of a better outcome [15]. The Jebsen−Taylor hand function test (JTT) was used to measure the performance speed of standardized tasks. The JTT consists of seven subtests, each scored on a scale of 0–15, with a higher score indicative of better hand function [16]. In the Perdue pegboard test (PPT), motor function is measured by the number of pins that can be placed in the board (along two parallel rows with 25 holes each) in 30 s, and dexterity is measured by the number of pins, washers, and collars that can be assembled in 60 s. Scores were evaluated for the affected right (dominant) hand and both hands [17]. Outcome measurements and data analyses were performed by a trained and blinded outcome assessor who was not involved in the intervention. Possible complications (pain, ecchymosis, skin abrasion, and swelling) were observed.

### 2.5. Statistical Analysis

Statistical analyses were performed using SPSS, version 23 (IBM Corp., Armonk, NY, USA). Fisher’s exact test was used to evaluate the homogeneity of the distribution of sex and burn types between the two groups before treatment, and the independent t-test was used to evaluate the homogeneity of distribution of the total burn surface area (TBSA), scar thickness, total VSS, dominant-hand and bilateral PPT scores, and the score for picking up small objects subtest of the JTT. A between-group *p*-value < 0.05 was deemed significant. The pre- to post-treatment scores were evaluated between the two groups using the Mann−Whitney or independent t-test, as appropriate for the variable type and distribution, with a *p*-value < 0.05 deemed significant. 

## 3. Results

There were no differences between the two groups in terms of demographic and clinical characteristic before treatment (*p* > 0.05 for all comparisons; Table 1). There were no differences in the measured outcomes before treatment between the two groups (*p* > 0.05 for all comparisons; Table 2). 

More reductions were found in the ESWT group than in the sham group for the pre- to post-treatment change in the VAS score (*p* = 0.001) and vascularity VSS score (*p* = 0.0015) (Table 3). There was a slight decrease in the pre- to post-treatment change of scar thickness (*p* = 0.018) for the ESWT group over the sham group (Table 3). However, there were no significant differences in the change score between the two groups for scar pigmentation (*p* = 0.19), pliability (*p* = 0.78), height (*p* = 0.66), and the total VSS score (*p* = 0.19; Table 3). Similarly, grip strength improved in both groups, without a difference in the change scores between the two groups (grasp, *p* = 0.99; lateral pinch, *p* = 0.46; and tip pinch, *p* = 0.26; Table 3). For the JTT, the change scores were significantly greater for the ESWT than the sham group for the subtasks of simulated card-turning (*p* = 0.02) and picking up small objects (*p* = 0.004), with no between-group differences in the change score for the remaining subtasks (writing, *p* = 0.12; stacking checkers, *p* = 0.15; simulated feeding, *p* = 0.99; picking up large light objects, *p* = 0.16; and picking up large heavy objects, *p* = 0.90; Table 3). For the PPT, there were no significant differences in the change scores for the affected hand (*p* = 0.12), both hands (*p* = 0.97), and assembly (*p* = 0.44; Table 3). Furthermore, there were no between-group differences in any of the subscores of the MHQ (Table 3): function (*p* = 0.17); activities of daily living, ADL (*p* = 0.51); work (*p* = 0.12); pain (*p* = 0.45); esthetics (*p* = 0.06); and satisfaction (*p* = 0.15). 

## 4. Discussion

We evaluated the effectiveness of ESWT on the hypertrophic scars of hand burns that required STSG. Therapeutic effects were evaluated by the change in scar pain, thickness, esthetic characteristics, and hand function after a four week period of treatment. The ESWT treatment of hypertrophic scarring after a burn injury to the hands provided significant benefits in improving hand function, decreasing pain, and suppressing hypertrophic scar growth. 

Cho et al. [12] have reported on the effects of ESWT applied to the area of most severe pain after complete epithelialization on pain outcomes (ESWT dose: EFD, 0.05 to 0.15 mJ/mm^2^; frequency, 4 Hz; and three sessions at one week intervals). The authors proposed that measured improvement in pain control with ESWT had resulted from a decrease in the number of unmyelinated nerve fibers; inhibition of nociceptors from repeated stimulation; regulation of neuroinflammatory molecules, such as substance P; and increased blood flow, facilitating tissue regeneration via an increase in endothelial nitric oxide synthase (eNOS). Joo et el. applied ESWT to burn scars with severe pruritus, using the same protocol as described by Cho et al., and found ESWT to be effective in reducing the severity of pruritus and improving ADLs [13]. Epithelial–mesenchymal transition (EMT), in which epithelial cells lose the inclination to reproduce, is known to have a role in fibrogenesis during hypertrophic scarring [18]. Seo et el. [1] identified that ESWT (EFD of 0.03, 0.1, and 0.3 mJ/mm^2^; a frequency of 4 Hz; and a volume of 1000 pulses) suppressed EMT by inhibiting the EMT inducer [1]. In our study, we used an intensity and frequency of ESWT previously shown to have a therapeutic effect for burns. Previous research has shown the effectiveness of low dose ESWT (0.1 mJ/mm^2^) in suppressing hypertrophic scar formation through inhibition of α-SMA expression [19,20]. Meirer et al. reported that ESWT, applied to partial-thickness burns, three and seven days after the burn injury (1500 impulses at 0.11 mJ/mm^2^), reduced the need for surgery and decreased scar formation during the six month follow-up period [21]. 

In our study, we also identified a significant benefit of ESWT in improving vascularization of the scar tissue, measured using the VSS. Previous research has investigated the mechanisms underlying the mechanosensitive feedback between ESWT and stimulated cells [22]. Nitric oxide (NO) production contributes to an increase in perfusion, with the improved blood supply preventing ischemia [22,23]. The accelerated epithelialization after ESWT can be explained by multiple factors, including neovascularization, increased granulation, and increased epithelial cell proliferation in the burn wound [8,9,24]. High-energy ESWT (0.3 mJ/mm^2^, frequency 4 Hz) increased the perfusion at remote sites via an increase of the NO and of vascular endothelial growth factor (VEGF) [25]. Davis et al. also demonstrated that the application of ESWT at 1 h postinjury significantly blunted the activity of inflammatory cells [26], although in clinical practice, hyperemia may occur at the site of ESWT due to an increase in local blood flow [6,13]. Overall, decreasing the healing time would be an important factor in reducing the risk of undesirable scarring and decreasing pain. 

ESWT is known to be effective for reducing pain and improving function on orthopedic injuries [27]. Substance P may play a role in pain mediation in small unmyelinated C fibers. An intradermal injection of substance P causes a flare-up in pain, swelling, and pruritus. On the other hand, calcitonin gene-related peptide (CGRP) is a well-known marker of unmyelinated C fibers and thinly myelinated Aδ fibers, both of which are important in nociception. The loss of nerve fibers and the depletion of neuropeptides with ESWT might be effective in decreasing pain. A low dose of ESWT (0.08 mJ/mm^2^, frequency 4 Hz) applied to a knee as a treatment for osteoarthritis reduced neuropeptide expression, which was associated with improved pain and gait function [28]. More substance P and CGRP are observed in painful hypertrophic scars compared to normal skin [29]. This evidence reinforces the idea that the pain-control effect of ESWT is likely to be mediated by its effect on pain mediators, and substance P and CGRP more specifically. The clinical benefits of ESWT have been reported for a range of orthopedic disorders and associated with self-reported improvements in function and quality of life [30,31,32]. 

ESWT has been widely used in treatment of various tendinopathies in upper limbs. Steroids limit tenocyte function by reducing proteoglycan and collagen synthesis, which reduce mechanical strength. Unlike steroid therapy, ESWT studies have shown that ESWT affects tenocyte regeneration, leading to an improvement in muscle strength. The effectiveness of ESWT compared with that of local steroid injections showed that ESWT had favorable effects on resolution of pain and improvement of grip strength [33,34]. In our study, we identified improvements in grip strength with ESWT, as well as improvement in simulated card-turning and the picking up of small objects. This improved function might be related to improvements in the pliability of the scar because of ESWT breaking down collagen fibers to induce scar remodeling. The increased pliability of the scars with ESWT, with a less evident effect on coloration, has already been reported [11]. A previous study [4] has also indicated that unfocused application of shock waves (EFD, 0.13 mJ/mm^2^; frequency, 6 Hz; two sessions per week for five weeks) was effective in significantly improving VSS scores, scarring pain, and the range of movement of the hand. These clinical benefits of ESWT were thought to be mediated by stimulation of dermal fibroblasts, neoangiogenesis, and improvement in the parallel organization of collagen fibers within the skin. Similarly, a beneficial effect of ESWT has been reported for the treatment of trigger finger complaints, resulting from enhanced repair and healing of the degenerated tendon [34]. Overall, as pain, pruritus, and decreased function negatively impact the quality of life after a burn, ESWT may have a global effect in improving quality of life after burn injuries [35]. 

This study requires cautious interpretation of the data, for reasons of small sample size, a short follow-up period, and the absence of detailed measurement of range of motion in the affected hand. Future studies with a longer time frame and more detailed assessment are needed to confirm our findings. Continued basic research into the mechanisms underlying the clinical effects of ESWT are needed to determine optimal parameters for the clinical management of hypertrophic scars.

## 5. Conclusions

In making the comparison of ESWT with a sham stimulation therapy for the treatment of hypertrophic scarring of the hand after a burn injury, we identified a clinically beneficial effect of ESWT in promoting hand function, improving scarring, and alleviating scar-related pain. This indicates the clinical usefulness of the intensity and frequency parameters of ESWT that we used in our study. It is important to point out that the optimal frequency and intensity of ESWT for the treatment of hypertrophic scars of the hands after a burn remain to be determined. Our findings, however, do provide experimenters with the option to use ESWT for its potential to improve the management and treatment of hand burn scars after STSG. 

## Figures and Tables

**Figure 1 jcm-09-01376-f001:**
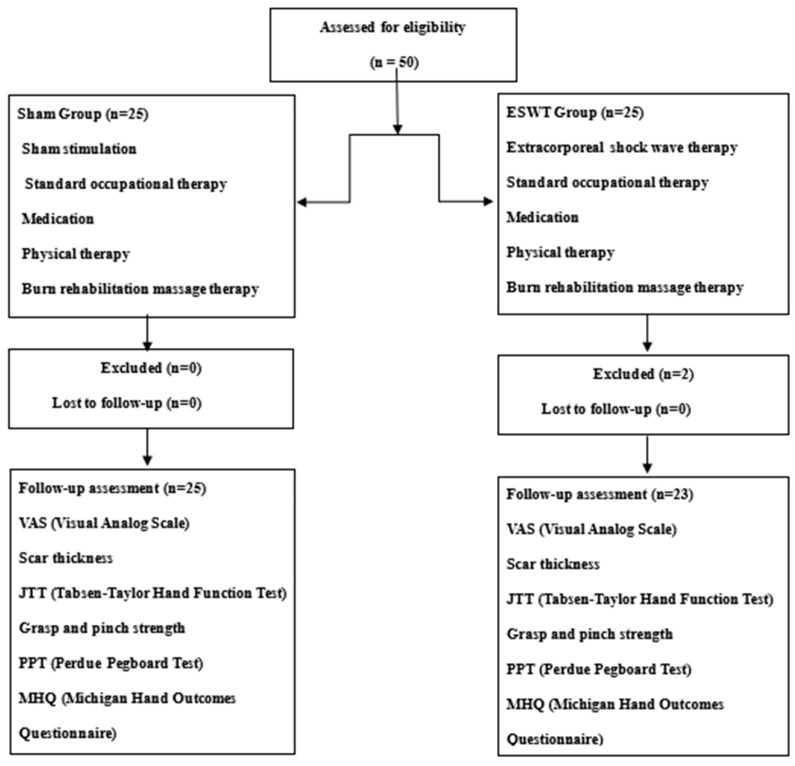
Diagram for subject enrollment, allocation, and follow-up.

**Figure 2 jcm-09-01376-f002:**
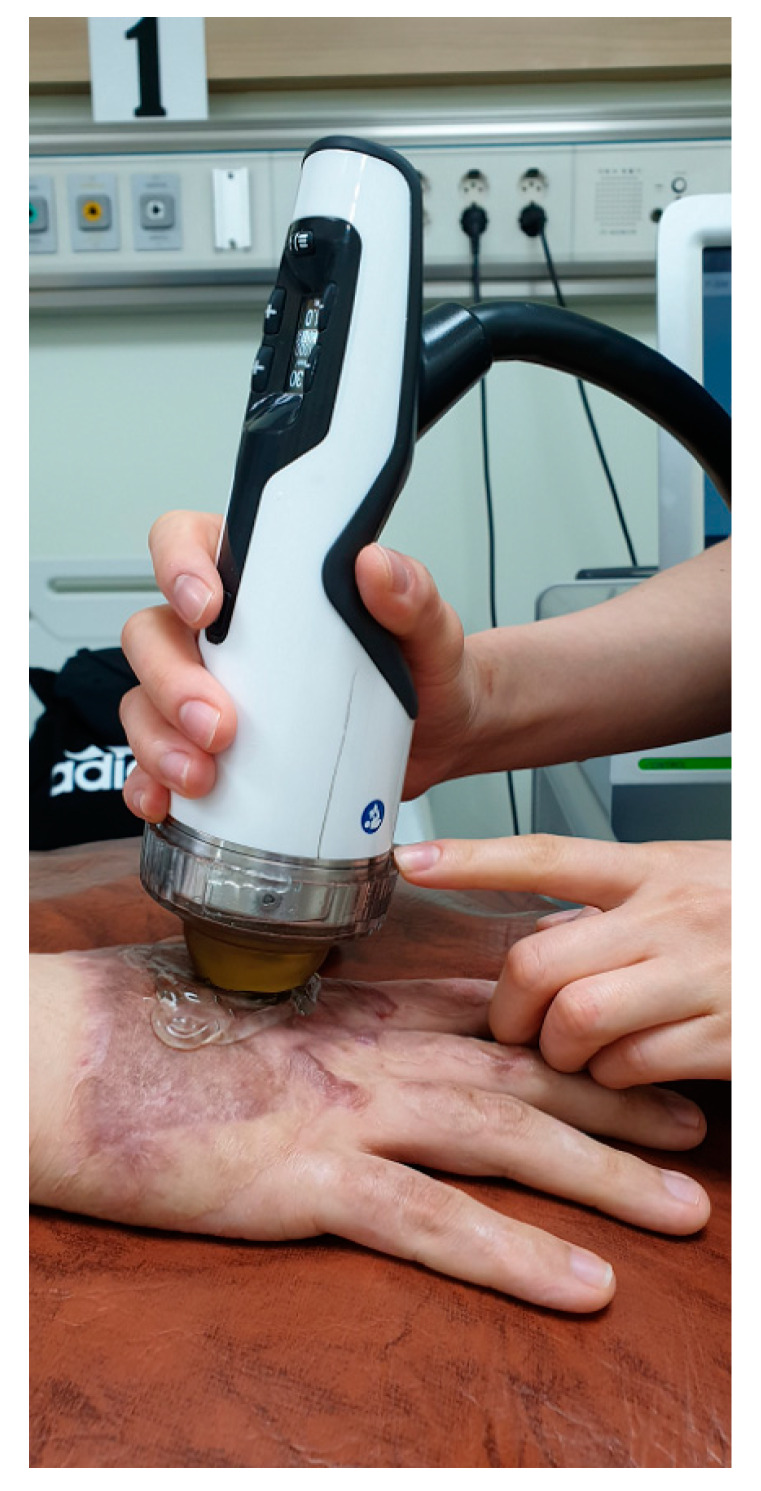
The extracorporeal shock wave therapy was administered to burn patients. The administered shock wave dose was 100 impulses/cm^2^ at 0.05 to 0.30 mJ/mm^2^ with a total of 1000 to 2000 impulses per treatment.

**Figure 3 jcm-09-01376-f003:**
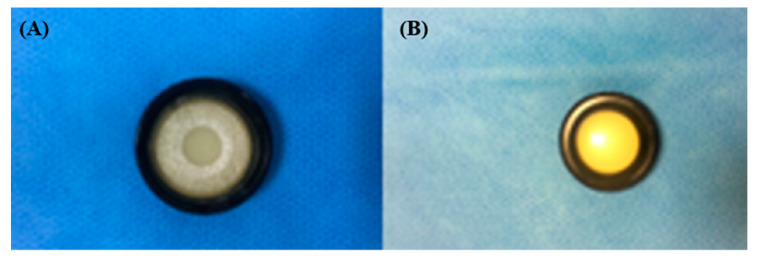
Sham adapter: (**A**) inside view; (**B**) outside view.

**Table 1 jcm-09-01376-t001:** Baseline characteristics of the study group.

	ESWT Group (n = 23)	Sham Group (n = 25)	*p-*Value
Male:Female	22:1	23:2	0.53
Age (years)	47.09 ± 11.09	48.56 ± 11.18	0.94
Cause of burn			0.43
Flame burn	15	18	
Electrical burn	1	3	
Contact burn	0	1	
Scalding burn	3	1	
Spark burn	4	2	
Time to treatment (days)	64.04 ± 36.62	63.48 ± 14.43	0.12
TBSA (%)	28.39 ± 17.86	27.80 ± 19.46	0.91

ESWT, Extracorporeal shock wave therapy; TBSA, total burn surface area. Values are presented as mean ± standard deviation. The *p*-values were calculated using Fisher’s exact test, the Mann−Whitney test, or Student’s t-test as appropriate.

**Table 2 jcm-09-01376-t002:** Outcome measures at baseline.

	ESWT Group (n = 23)	Sham Group (n = 25)	*p*-Value
**VAS**	7.00 ± 1.24	6.96 ± 1.21	0.84
**Thickness (cm)**	0.21 ± 0.11	0.19 ± 0.06	0.36
**Vancouver Scar Scale**			
Pigmentation	3.00 ± 0.00	2.92 ± 0.28	0.17
Pliability	1.87 ± 0.76	2.20 ± 0.76	0.14
Height	1.43 ± 0.51	1.56 ± 0.58	0.33
Vascularity	2.17 ± 0.72	2.32 ± 0.75	0.45
Total	8.48 ± 1.70	9.00 ± 1.85	0.32
**Grasp and Pinch Power Test**			
Grasp (kg)	7.50 ± 7.40	5.81 ± 7.72	0.38
Lateral Pinch (kg)	3.21 ± 1.67	3.58 ± 5.21	0.43
Tip Pinch (kg)	1.40 ± 0.91	1.32 ± 1.33	0.56
**Jebsen−Taylor Hand Function Test**			
Writing	11.04 ± 4.80	10.64 ± 3.49	0.10
Cards	3.57 ± 2.78	3.04 ± 2.42	0.53
Small	5.57 ± 3.86	5.04 ± 4.23	0.66
Checkers	9.26 ± 4.11	8.00 ± 5.38	0.72
Feeding	10.09 ± 4.07	8.88 ± 5.23	0.73
Light	8.78 ± 4.52	9.68 ± 3.69	0.60
Heavy	8.52 ± 4.33	8.08 ± 4.47	0.76
**Perdue Pegboard Test**			
Affected hand	7.78 ± 5.51	8.52 ± 3.37	0.58
Both hands	5.78 ± 4.49	6.36 ± 2.61	0.59
Assembly	12.13 ± 12.33	14.00 ± 8.45	0.26
**Michigan Hand Outcomes Questionnaire**			
Function	21.09 ± 12.96	23.80 ± 10.54	0.30
ADL	21.52 ± 12.38	23.20 ± 12.74	0.49
Work	15.57 ± 14.24	15.20 ± 18.06	0.63
Pain	34.35 ± 28.13	24.92 ± 21.27	0.22
Esthetics	12.36 ± 16.36	10.08 ± 13.78	0.74
Satisfaction	24.96 ± 12.35	25.08 ± 14.10	0.75

VAS, visual analogue scale; ESWT, extracorporeal shock wave therapy. Values are presented as the mean ± standard deviation. The *p*-values were calculated using Fisher’s exact test, the Mann−Whitney test, or Student’s t-test as appropriate.

**Table 3 jcm-09-01376-t003:** Change score (pre- to post-treatment) on measured outcomes.

	ESWT Group (n = 23)	Sham Group (n = 25)	*p*-Value
**VAS**	−1.48 ± 1.04	−0.52 ± 0.77	0.001 **
**Scar thickness (cm)**	0.01 ± 0.07	0.07 ± 0.07	0.018 **
**Vancouver Scar Scale**			
Pigmentation	−0.04 ± 0.21	−0.16 ± 0.37	0.19
Pliability	0.13 ± 0.81	0.08 ± 1.08	0.78
Height	0.09 ± 0.60	0.44 ± 0.58	0.06
Vascularity	−0.52 ± 0.73	−0.04 ± 0.61	0.015 **
Total	−0.35 ± 1.61	0.32 ± 1.75	0.19
**Grasp and Pinch Power Test**			
Grasp (kg)	4.71 ± 4.90	4.71 ± 3.77	0.99
Lateral Pinch (kg)	1.10 ± 1.69	0.74 ± 1.61	0.46
Tip Pinch (kg)	0.80 ± 1.13	0.48 ± 0.78	0.26
**Jebsen−Taylor Hand Function Test**			
Writing	1.57 ± 4.11	0.88 ± 3.43	0.12
Cards	2.48 ± 3.54	0.40 ± 2.22	0.02 *
Small	3.70 ± 4.30	0.16 ± 2.87	0.004 **
Checkers	1.30 ± 4.48	3.32 ± 4.40	0.15
Feeding	1.70 ± 3.97	1.48 ± 2.38	0.99
Light	1.78 ± 4.77	0.04 ± 2.17	0.16
Heavy	1.39 ± 5.06	0.60 ± 1.68	0.90
**Perdue Pegboard Test**			
Affected hand	2.26 ± 3.51	0.00 ± 3.65	0.12
Both hands	1.61 ± 2.97	1.44 ± 4.23	0.97
Assembly	4.04 ± 10.55	1.72 ± 10.15	0.44
**Michigan Hand Outcomes Questionnaire**			
Function	13.78 ± 18.12	8.60 ± 15.65	0.17
ADL	11.74 ± 20.43	12.20 ± 20.77	0.51
Work	19.22 ± 19.47	13.72 ± 22.91	0.12
Pain	−9.39 ± 24.78	−6.48 ± 18.45	0.45
Esthetics	10.55 ± 16.46	9.60 ± 21.33	0.15
Satisfaction	22.86 ± 21.59	10.30 ± 19.37	0.06

VAS, visual analogue scale; ESWT, extracorporeal shock wave therapy. Values are presented as the mean ± standard deviation. The *p*-values for between-group differences were calculated using the Mann−Whitney test (**, *p* < 0.05) and independent t-test (*, *p* < 0.05), as appropriate.

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
