# Peer review of "Clinical Utility of Extracorporeal Shock Wave Therapy on Hypertrophic Scars of the Hand Caused by Burn Injury: A Prospective, Randomized, Double-Blinded Study"

_jcm, 2020, doi:10.3390/jcm9051376_

Round 1

Reviewer 1 Report

This is an interesting study by Joo et al. I agree that is a small sample size with short period of follow-up. A preliminary data on hand range of motion between the groups in this manuscript will help to complement the manuscript. 

Author Response

Answer> We appreciate you careful advise. For detailed information, the hand ROM parameter could be analyzed. However the hand ROM parameter is not included in routine hand function evaluations. We add the limitations of this study in the discussion section. In the future, a study on the effect of ESWT on hypertrophic scars on hands will evaluate the range of joint motion.

We changed the title of this study from “Clinical utility of extracorporeal shock wave therapy for the treatment of hand burns” to “Clinical utility of extracorporeal shock wave therapy on hypertrophic scars of the hand with burn injury”. We hope that the tile change will be correctly understood by the reader.

Reviewer 2 Report

The authors review the effects of ESWT on hand burns. I have concerns about the conclusions of the paper. If you look at the data, you show improvement with change of VAS but the treatment started with a much higher score to start with so they would have a higher chance of having more change. When I look at the scar thickness, it appears that there is greater improvement with controls; but the authors conclude the opposite. Your data does not support that finding. Otherwise, there is no difference, except for vascularity, which always improves.
Therefore, I do not find the evidence to support your findings.

1) The title is really not accurate - you are not treating hand burns, you are treating "hypertrophic scars resulting from hand burns". The current title suggests that you are treating the open wounds but you are really treating their scars. You state that the burns were in the "re-epithelialization phase of wound healing" but it would be unlikely to have a hand burn not closed months after injury. 

2) You claim that left-hand dominance is an exclusion from the trial but don't you really mean "non-dominant" hand? 

3) Are the evaluators of scarring and range of motion blinded to the treatment?

4) It would be helpful to differentiate those hand that were skin grafted versus those that healed on their own. 

5) Can you explain how grip strength is improved by ESWT?

Author Response

1. If you look at the data, you show improvement with change of VAS but the treatment started with a much higher score to start with so they would have a higher chance of having more change.

Answer> We appreciate you careful advise. The results of the sham group were incorrectly written for the VAS score at baseline in Table 2, and the statistical results were checked again.

2. When I look at the scar thickness, it appears that there is greater improvement with controls; but the authors conclude the opposite. Your data does not support that finding.

Answer> We appreciate you careful advise. The thickness of hypertrophic scars in the ESWT group increased less than the sham group to a statistically significant level. The result section were revised so that the reader could understand clearly.

 3. The title is really not accurate - you are not treating hand burns, you are treating "hypertrophic scars resulting from hand burns". The current title suggests that you are treating the open wounds but you are really treating their scars. You state that the burns were in the "re-epithelialization phase of wound healing" but it would be unlikely to have a hand burn not closed months after injury. 

Answer> We agree with the reviewer. We changed the title of this study from “Clinical utility of extracorporeal shock wave therapy for the treatment of hand burns” to “Clinical utility of extracorporeal shock wave therapy on hypertrophic scars of the hand with burn injury”. We hope that the tile change will be correctly understood by the reader.

 4. You claim that left-hand dominance is an exclusion from the trial but don't you really mean "non-dominant" hand? 

Answer> We appreciate you careful advise. We included only the right hand, which had undergone a split-thickness skin graft (STSG), among the right-handed before the thermal injury. We have added the descriptions of inclusion criterias in method section.

 5. Are the evaluators of scarring and range of motion blinded to the treatment?

Answer> We agree with the reviewer. Outcome measurements and data analyses were performed by a trained and blinded outcome assessor who was not involved in the intervention. We have added a more detailed descriptions in the method section.

 6. It would be helpful to differentiate those hand that were skin grafted versus those that healed on their own. 

Answer> To evaluate the effectiveness of ESWT accurately, the comparison was limited to the right hand, and the scars which undergone STSG. We added the descriptions of inclusion criterias in method section.

 7. Can you explain how grip strength is improved by ESWT?

Answer> We appreciate you careful advise. In this study, grip strength improved in both groups, without a difference in the change score between the two groups (grasp, p=0.99; lateral pinch, p=0.46; and tip pinch, p=0.26). ESWT studies on tendinopathies have shown that ESWT affects tenocytes regeneration, leading to resolution of pain and improvement of grip strength. We add the additional references in the discussion section.

Round 2

Reviewer 2 Report

Accept